# Engineered Cross-Linked Silane with Urea Polymer Thin Durable Coatings onto Polymeric Films for Controlled Antiviral Release of Activated Chlorine and Essential Oils

**DOI:** 10.3390/jfb14050270

**Published:** 2023-05-12

**Authors:** Elisheva Sasson, Omer Agazani, Eyal Malka, Meital Reches, Shlomo Margel

**Affiliations:** 1Bar-Ilan Institute of Nanotechnology and Advanced Materials (BINA) and Department of Chemistry, Bar-Ilan University, Ramat-Gan 5290002, Israel; elisheva.sa@gmail.com (E.S.);; 2Institute of Chemistry and the Center for Nanoscience and Nanotechnology, The Hebrew University of Jerusalem, Jerusalem 9190401, Israel

**Keywords:** COVID-19, antiviral coatings, N-halamine compounds, T4 bacteriophage, essential oils, canine coronavirus

## Abstract

In March 2020, the World Health Organization announced a pandemic attributed to SARS-CoV-2, a novel beta-coronavirus, which spread widely from China. As a result, the need for antiviral surfaces has increased significantly. Here, the preparation and characterization of new antiviral coatings on polycarbonate (PC) for controlled release of activated chlorine (Cl^+^) and thymol separately and combined are described. Thin coatings were prepared by polymerization of 1-[3-(trimethoxysilyl)propyl] urea (TMSPU) in ethanol/water basic solution by modified Stöber polymerization, followed by spreading the formed dispersion onto surface-oxidized PC film using a Mayer rod with appropriate thickness. Activated Cl-releasing coating was prepared by chlorination of the PC/SiO_2_-urea film with NaOCl through the urea amide groups to form a Cl-amine derivatized coating. Thymol releasing coating was prepared by linking thymol to TMSPU or its polymer via hydrogen bonds between thymol hydroxyl and urea amide groups. The activity towards T4 bacteriophage and canine coronavirus (CCV) was measured. PC/SiO_2_-urea-thymol enhanced bacteriophage persistence, while PC/SiO_2_-urea-Cl reduced its amount by 84%. Temperature-dependent release is presented. Surprisingly, the combination of thymol and chlorine had an improved antiviral activity, reducing the amount of both viruses by four orders of magnitude, indicating synergistic activity. For CCV, coating with only thymol was inactive, while SiO_2_-urea-Cl reduced it below a detectable level.

## 1. Introduction

COVID-19, the pandemic attributed to a novel coronavirus, had a drastic effect worldwide, with over 4.3 million confirmed deaths at the initial stage in 2020 [1]. The persistence of SARS-CoV-2 on surfaces triggered a focus on strategies that decrease their viral contamination. The virus is infectious for up to 8 h on cardboard and copper, and 72 h on steel/plastic surfaces [2,3]. Other human coronaviruses are infectious for 9 days [4,5]. Antiviral surfaces and coatings have been explored to address this phenomenon. Between 2020 and 2022, 512 papers were published regarding polymers and COVID-19 [6]. Despite great efforts to develop effective antiviral drugs against SARS-CoV-2, approval by the Food and Drug Administration (FDA) has been quite limited [7]. However, over the years, most reports focus on anti-bacterial abilities, while there are fewer reports on antiviral functionality, coatings or surfaces. The reason for this is that in the absence of host cells, some viruses immediately become inactive [5,8]. Moreover, some viruses are not able to spread without a host [5,9]. SARS-CoV-2, however, persists on surfaces, and therefore has the potential to be transmitted by surfaces [4,10].

Reports on antiviral surfaces include a comparison between copper and stainless steel surfaces. Noyce et al. showed that copper surfaces can reduce the amount of Influenza A by three orders of magnitude compared to stainless steel [11]. Nanoparticles and ions of copper also show antiviral activity [12,13]. Other inorganic materials such as silver and zinc are also active [14,15,16,17,18]. Moreover, organic compounds such as polycations are used in antiviral coatings [5,19,20,21].

Interest in compounds with covalent bonds between nitrogen and halogen is growing, owing to their excellent antimicrobial ability. Moreover, such N-halamines exhibit prolonged stability in aqueous solution, weak toxicity and low cost, and can be readily regenerated [22,23,24]. Such compounds are attractive for disinfection of water and food [25]. Their aqueous dissociation constant is quite low, and increases from imide to amide to amine; however, oxidative transfer of chlorine to microorganisms rather than hydrolysis is significantly favored (Figure 1). Although the hydrolysis constant is around 10^−8^, the transfer of oxidative chlorine from amide-halamine compounds to the bacteria is much more favorable [26,27,28]. For industrial applications, such compounds are considered the most practical due to their moderate chlorine transfer rate.

Thymol is one of the main components in thyme, which is widely used in pharmacology [29]. This molecule has anti-bacterial, anti-fungal, and some antiviral activities [29,30,31,32,33]. There are many examples of anti-bacterial and anti-fungal surfaces that contain thymol [34,35,36,37] albeit only a few reports on thymol as an antiviral agent [31,38,39]. Thymol can inactivate herpes from infecting cells, probably by damaging the virion envelope structure required for adsorption/entry [40]. Moreover, Castanas et al. showed that essential oils mixture (mainly menthol, 1,8-cienol and limonene) can reduce the release of the delta variant pseudo virus of SARS-CoV-2 by more than 80% [41,42]. Other studies have shown the antiviral activity of other mixtures of essential oils, including thymol, against SARS-CoV-2 [42,43,44,45]. It is worth mentioning that those studies report on the antiviral activity of essential oil mixtures and not on their ability to serve as antiviral coatings.

An in silico study on the transmembrane protease serine 2 (TMPRSS2) suggests that thymol can inhibit this enzyme’s host, which plays a major role in the entry of SARS-CoV-2 [46]. Despite all of the above, we are not aware of any report on antiviral coatings based on essential oils. Here, we report a new antiviral coating on PC film composed of SiO_2_-urea that may bind to activated Cl and/or thymol and allow their controlled release, both separately and together, upon contact with a virus. Thin SiO_2_-urea thin coatings on PC film were prepared by modified Stöber polymerization of 1-[3-(trimethoxysilyl)propyl] urea (TMSPU) in an ethanol/H_2_O continuous phase, as shown in Figure 2. 

Chlorine-releasing coatings were prepared by chlorination of the SiO_2_-urea by NaOCl via the urea amide groups to obtain N-chloramine groups, as shown in Figure 2 [34]. Thymol-releasing coating was prepared by adding the thymol (Figure 3) to the silane-urea monomer (TMSPU, Figure 3), followed by Stöber polymerization in presence of the corona-treated PC film, as shown in Figure 2. The antiviral activity of SiO_2_-urea-Cl, SiO_2_-urea-thymol and SiO_2_-urea-Cl-thymol coatings on the PC films was measured against two different viruses: T4 bacteriophage and canine coronavirus (CCV). This covering can be used as a stable coating for protecting partitions against pathogenic viruses, including SARS-CoV-2.

## 2. Experimental

### 2.1. Materials

The following analytical-grade chemicals were purchased from Sigma-Aldrich (St. Louis, MO, USA) and used without further purification: ethanol (AR grade), ammonium hydroxide (NH_4_OH, 28%), 1-[3-(trimethoxysilyl)propyl] urea (TMSPU), sodium hypochlorite (NaOCl) aqueous solution (5%) and thymol oil. Double-distilled water (DDW) and triple-distilled water (TDW) were obtained from a TREION™ purification system (Tel-Aviv, Israel). Corona-treated PC films were provided by Plaskolite (Gazit, Tel Aviv, Israel).

*Escherichia coli* strain B (Migula), Castellani and Chalmers (ATCC 11303) and *E. coli* bacteriophage T4 (ATCC 11303-B4) were obtained from the ATCC (Manassas, VA, USA). Lysogeny broth (LB) and agar were, respectively acquired from Merck (Rahway, NJ, USA) and Becton Dickinson (Franklin Lakes, NJ, USA). Crandell Rees feline kidney (CRFK) cells (ATCC CCL94), CCV (VR2068), L-alanyl-L-glutamine (200 mM), medium with essential minimum Earl salt (EMEM), penicillin−streptomycin−amphotericin (PSA), trypsin−EDTA, donor horse serum (DHS) and Dulbecco’s phosphate-buffered saline (DPBS) were obtained from Biological Industries (Beit-HaEmek, Israel). Trypsin 1:250 was acquired from Bio-World (Dublin, OH, USA). 

### 2.2. Methodology

#### 2.2.1. Preparation of SiO_2_-Urea and SiO_2_-Urea-Thymol Coatings on PC Film

The SiO_2_-urea coating on PC film was prepared by modified Stöber polymerization of TMSPU in continuous ethanol/water phase. In a typical experiment, surface oxidation of PC films was carried out by air corona treatment (450 W·min/m^2^) [47]. Ethanol (18.75 mL), water (3 mL), ammonium hydroxide (0.675 mL) and TMSPU (1.5 mL) in the absence or presence of thymol (1% *w*/*v*, 0.25 g) were added to 50 mL tubes and mixed at room temperature (RT) for about half an hour. The corona-treated PC films were then coated by a Mayer rod with the formed coating dispersion. 

#### 2.2.2. Rod-Coating Technique

The Corona-treated PC film was coated with TMSPU by spreading the basic mixture of TMSPU in ethanol/water in the absence or presence of thymol using a Mayer rod obtained from RK Print Coat Instruments Ltd. (Litlington, UK) [16,17]; this is a metal bar wrapped in a wire, which draws a solution over the surface of the substrate (Figure 4). The film thickness is determined by the wire diameter. The substrate was dried for a few minutes in an oven at 110 °C. The formed PC/SiO_2_-urea and PC/SiO_2_-urea-thymol films were washed (with water) and dried in air. 

Rods are available in a wide variety of wire sizes to provide a range of coating thickness. The concentration of TMSPU in the coating solution determines the dry thickness. In this study, uniform layers with a wet thickness of 400, 200 and 120 µm were prepared. For biological studies, only coatings with a wet thickness of 400 µm were used.

#### 2.2.3. SiO_2_-Urea-Cl Coating on PC Film

##### Chlorination of the SiO_2_-Urea Coating

After completing the preparation of the SiO_2_-urea coating on the PC film, activated chlorine binding to the amide groups of the urea was carried out by the following chlorination process. Sodium hypochlorite aqueous solution (0.5% *w*/*v*) was put in a box that contained the coated film. The box was then shaken at 40 °C for 12 h, followed by removal of excess sodium hypochlorite through extensive washing with water.

##### Active Chlorine Determination

Iodometric sodium thiosulfate titration was used to determine the content of chlorine that was bound [48,49]:$$\left[Cl^{+}\right]\;\left(mM\right)=\frac{1000VN}{2}$$
where *V* is the volume and *N* is the normality of the titrated solution.

#### 2.2.4. Characterization of the Coatings

##### Atomic Force Microscope 

Atomic force microscopy (AFM) measurements of the films were carried out using a scanning probe Bio FastScan microscope (Bruker AXS), with quantitative nanomechanical mapping (QNM) Peak Force mode using a silicon probe (FastScan-C, Bruker, Billerica, MA, USA) with 0.45 N/m spring constant. An acoustic hood was used for minimal vibrational noise, with a scan rate of 1.7 Hz in the retrace direction. The resolution was 512 samples per line. Nanoscope Analysis software (v1.5) was used for image processing and thickness analysis, applying the “flatting” and “planefit” functions. Three areas of the film were averaged (three measurements per area).

##### Fourier Transform Infrared Spectroscopy

Fourier transform infrared spectroscopy (FTIR) measurements used the attenuated total reflectance (ATR) technique with ALPHA-FTIR (Bruker) QuickSnap sampling with a platinum diamond ATR module.

##### Contact Angles

Water contact angle (CA) sessile drop measurements were made with a goniometer OCA20 system from Data Physics Instruments Gmbh (Fielderstadt, Germany). Drops of DDW (5 µL) were placed on three areas of the film, and images were captured after a few seconds. Laplace–Young curve fitting was used to obtain static CA values. All measurements were performed under the same conditions.

##### X-ray Photoelectron Spectroscopy

X-ray photoelectron spectroscopy (XPS) measurements were made using a Nexsa (Bristol, UK) spectrometer with a monochromated micro-focused Al Kα source (with a low photon energy of 1486.6 eV). Survey/high-resolution spectra were acquired at pass energy of 200/50 eV. The usual source power was 72 W. Binding energies were recalibrated with CC/CH of C 1s set at 285 eV. UHV conditions were applied with a base pressure of 5 × 10^−10^ torr (up to 3 × 10^−9^ torr). Analysis and deconvolution were performed using software by Vision 2 (Kratos, UK). 

#### 2.2.5. Release Rates at Different Temperatures of Activated Cl from PC/SiO_2_-Urea-Cl Films 

PC/SiO_2_-urea-Cl and PC/SiO_2_-urea-Cl-thymol films (1 cm^2^ pieces) were exposed at RT, 4 and −4 °C for 0, 1, 3, 4, 24, 48, 72, 96 and 500 h, during which time Cl was released from the coatings. After the set exposure time, the films were tested by iodometric/thiosulfate titration to quantify the concentration of Cl^+^ still present on the coatings. 

#### 2.2.6. Determination of Thymol Content in PC/SiO_2_-Urea Thymol-Containing Films

PC/SiO_2_-urea-Cl and PC/SiO_2_-urea-Cl-thymol films (4 cm^2^ pieces) in 20 mL ethanol were centrifuged (6500 rpm, 35 min) until complete thymol extraction (further centrifugation indicated no remnants of trapped thymol). The content of thymol was obtained using a pre-calibrated curve (max absorption at λ = 276 nm) with a UV-visible spectrophotometer (CARY 1E, Varian, Palo Alto, CA, USA). The extracted thymol concentration was derived from the calibration curve (Appendix A). Measurements were made in triplicate. 

#### 2.2.7. Release Rates of Thymol from PC/SiO_2_-Urea-Thymol and PC/SiO_2_-Urea-Cl-Thymol Films 

PC/SiO_2_ urea-thymol and PC/SiO_2_-urea-Cl-thymol coated films were cut to 4 cm^2^ pieces and left exposed for 0, 4, 8, 22 and 264 h at RT, during which time thymol vapors were released. After each exposure time, the sample in ethanol was centrifuged (6500 rpm, 35 min) until complete thymol extraction. The release rate was determined by subtracting the amount of thymol in the ethanol extract from the initial amount. Measurements were made in triplicate.

#### 2.2.8. Durability of SiO_2_-Urea Coating on PC Film

The strength of the interaction between the SiO_2_-urea coating and the film was tested. An adhesive tape was pressed firmly on the film, then slowly peeled off (Appendix A) [50]. The process was repeated 25 times, and FTIR spectra were measured periodically. The coating’s stability in water was evaluated by soaking for six months and drying.

#### 2.2.9. Bacteriophage Assay

The *E. coli* bacteriophage T4 stock solution was diluted to 1.0 × 10^6^ PFU/mL. 16 µL drops were incubated at RT on the bare and coated PC films for 24 h with Parafilm on top of the drop. The size of the films and the Parafilm were 1 cm × 1 cm. The incubation was carried out in a humid environment by adding 2 mL of DDW near each sample. After incubation, soya casein digest lecithin polysorbate (SCDLP, 2 mL) was added and the samples were shaken (15 min, 120 rpm) to remove the phages from the surface to the solution. The solution was diluted with Luria–Bertani (LB, 8 mg/mL) solution to a 10-fold dilution series. The T4 and *E. coli* starter solution were mixed in 0.6% agarose solution and spread on plates with agar incubated at 37 °C overnight to form plaques. The plaques were manually quantified and normalized with respect to the bare polycarbonate.

#### 2.2.10. CCV Assay

Some 16 µL of diluted T4 solution was incubated as described above using TDW instead of DDW. The viruses were removed from the surface by adding 2 mL of TDW. A 10-fold dilution series in EMEM was prepared with L-alanyl-L-glutamine (2 mM) PSA (1% solution), and the TCID_50_/mL was measured. Six surfaces were included (three coated/uncoated). In a 96-well plate, CRFK cells (10,000/well) were incubated to 80–90% confluence and washed with DPBS (0.2 mL/well). The diluted CCV solution was added (0.1 mL/well) and the plate was incubated (37 °C, 5% CO_2_, 5–6 repeats). After 2 h, EMEM (2 mM L-Glu, 1% PSA), DHS (1% final concentration), and trypsin (1:250, 1 μg/mL final concentration) were added (total volume 0.2 mL/well). The plates were incubated for 5–7 days until a cytopathic effect was observed using a Zeiss Primovert inverted microscope (Carl Zeiss AG, Jena, Germany). Based on the number of wells with an observed cytopathic effect, TCID50/mL values were calculated according to the improved Kärber method [21]. Micrographs were taken using Axiocam ICC3 (Carl Zeiss AG, Jena, Germany).

## 3. Results and Discussion

### 3.1. Characterization of the Coatings

#### 3.1.1. Atomic Force Microscopy (AFM)

Air corona-treated PC films coated with SiO_2_-urea, SiO_2_-urea-Cl, SiO_2_-urea-thymol and SiO_2_-urea-Cl-thymol were measured for their surface roughness using AFM (Table 1, Figure 5). The treated films had much higher roughness: 11.7 ± 0.5 compared to only 0.40 ± 0.02 nm for the non-treated film, due to pores created by collisions with oxygen ions. After coating with SiO_2_-urea and SiO_2_-urea-thymol, Table 1 shows a pronounced decrease in roughness (0.85 ± 0.01 and 1.2 ± 0.1 nm, respectively) possibly due to filling of the pores achieved by the corona treatment. However, coating the PC/SiO_2_-urea with activated chlorine increased the roughness considerably from 0.85 ± 0.01 to 6.7 ± 0.3 nm due to the use of a strong oxidizer (NaOCl) on the coated surface. Thymol increased the roughness further to 9.1 ± 0.7 nm due to combination with the active chlorine.

#### 3.1.2. Fourier Transform IR/Attenuated Total Reflection

FTIR/ATR was used to verify TMSPU polymerization on the corona-treated PC films, followed by chlorination and/or thymol linking to the SiO_2_-urea coating. Figure 6 presents spectra of the corona-treated PC and PC/SiO_2_-urea films before and after chlorination (top). The principal peaks of the PC spectrum [51] are shown in Figure 6 (bottom): aromatic C–H deformations (~3000 cm^−1^), C=O deformations (~1775 cm^−1^), C=C vibrations (1506 cm^−1^), asymmetric O–C–O carbonate deformations (1232–1164 cm^−1^) and CH_3_ vibrations (2800–3000 cm^−1^). The PC/SiO_2_-urea film spectrum (Figure 6, middle) includes Si-O-Si and SiOH deformations around 1132 and 1041 cm^−1^, and deformations of the urea N-H vibrations (3348 cm^−1^). The PC/SiO_2_-urea-Cl spectrum (Figure 6, top) displays the same peaks as the PC/SiO_2_-urea film, except for the urea peak at 3348 cm^−1^, which decreases, and an additional peak (1673 cm^−1^) corresponding to the new Cl-activated N–Cl bond [52].

Figure 7 presents the spectra of thymol, PC/SiO_2_-urea-thymol and PC/SiO_2_-urea-Cl-thymol. The thymol spectrum [53] includes (bottom of Figure 7) peaks of hydroxyl (OH) around 3323 cm^−1^, CH_3_ stretching at 2957–2930 cm^−1^, phenyl (1454 cm^−1^), and phenol C-O stretching and in-plane OH bending (1344 and 1380 cm^−1^, respectively). At 806 cm^−1^, the peak is ascribed to the aromatic out-of-plane C-H and the ring at 947–806 cm^−1^. The PC/SiO_2_-urea-thymol film spectrum is presented in the middle of Figure 7. This spectrum shows that thymol is present in the PC/silica-urea film, as indicated by CH_3_ stretching (2930–2853 cm^−1^), the phenyl ring (1454, 930–782 cm^−1^), and aromatic out-of-plane C-H wagging vibrations (806 cm^−1^). The PC/SiO_2_-urea-Cl-thymol spectrum presented at the top of Figure 7 illustrates the presence of both thymol and activated chlorine.

#### 3.1.3. Contact Angle (CA)

Water CAs were determined to evaluate the hydrophilicity/hydrophobicity of the different thin coatings. Table 2 shows, as expected, that corona treatment decreased the CA from 74 ± 2° to 40 ± 2°, probably due to surface oxidation. Thin coating of the treated PC film with SiO_2_-urea, followed by chlorination or linking of thymol, lead to significant increase to 93 ± 4° and 110 ± 6°, respectively, due to the hydrophobic nature of Cl and thymol.

### 3.2. Determination of Cl Content in PC/SiO_2_-Urea-Cl Film

Table 3 illustrates a direct connection between the thickness of the SiO_2_-urea coating and the Cl concentration bonded to the SiO_2_-urea coating. Higher coating thickness leads to increased bound Cl. Increasing the wet thickness (as determined from the Mayer rod) of the SiO_2_-urea-Cl onto PC films from 120 to 200/400 µm increased the Cl content from 0.75 to 1.5/3.5 µmol/cm^2^. 

#### 3.2.1. X-ray Photoelectron Spectroscopy

The surface composition of the films was measured using XPS (Figure 8). The characteristic peaks of N 1s (400.02 eV) and Si 2p (102.1 eV) in Figure 8B,C, respectively, indicate the presence of the Si-urea coating in both PC/SiO_2_-urea (orange) and PC/SiO_2_-urea-Cl (gray) films. Figure 8D exhibits the success of the chlorination of the SiO_2_-urea coating by the Cl 2p peak at 200 eV. PC/SiO_2_-urea-thymol films were not tested, as thymol can evaporate during XPS.

#### 3.2.2. Release Rates of Chlorine from PC/SiO_2_-Urea-Cl and PC/SiO_2_-Urea-Cl-Thymol Films

The Section 2 describes the activated chlorine analysis and the evaluation of Cl^+^ release rates from the PC/SiO_2_-urea-Cl and PC/SiO_2_-urea-Cl-thymol films. Figure 9 exhibits release rates from PC/SiO_2_-urea-Cl at RT, 4 °C and −4 °C. In the first 24 h, 20% of the Cl^+^ content is released; after 48 h, 70% is released, which increases to 90% after a few weeks (500 h). Storing the coating at 4 °C (green line) significantly slows down the release rate, as shown in Figure 9. In addition, storing the coating at −4 °C does not release any chlorine even after 500 h. This is probably due to the significant decrease in exposure to microorganisms at these temperatures. Similar release rates of Cl^+^ as a function of temperature were also observed for the PC/SiO_2_-urea-Cl-thymol film.

### 3.3. Determination of Thymol Content in PC/SiO_2_-Urea Films

Thymol bound to films with and without added Cl was determined as described in the Section 2. PC/SiO_2_-urea-thymol and PC/SiO_2_-urea-Cl-thymol films exhibited similar binding: 933 ± 81 µg and 924 ± 12 µg of thymol per 4 cm^2^, respectively.

#### Release Rates of Thymol from PC/SiO_2_-Urea-Thymol and PC/SiO_2_-Urea-Cl-Thymol Films

Figure 10 exhibits the release rates of thymol at room and low temperatures (−4 °C). At RT, both coatings exhibit a massive release of thymol after 10 h, after which the rate decreases. After 22 h, about 60% of the thymol was released from PC/SiO_2_-urea-thymol. However, the release from PC/SiO_2_-urea-Cl-thymol is considerably slower (~25%). This is in accordance with hydrophobic interactions between the aromatic ring of thymol and the chlorine of the N-halamine, which are expected to decrease the release rate. Figure 10 also shows that storing both films at −4 °C completely prevents the release of thymol, a behavior similar to that illustrated in Figure 9 for the release of activated Cl from the PC/SiO_2_-urea-Cl film.

### 3.4. Durability of SiO_2_-Urea Coating on PC Film

SiO_2_-urea coating durability was examined as described in the Section 2. Similar results were obtained in FTIR after the tape and soaking tests (Appendix A, respectively). Hence, the PC/SiO_2_-urea coatings are stable and durable in water for six months. 

### 3.5. Antiviral Activity

The coatings were tested towards two different viruses: T4 bacteriophage and canine coronavirus (CCV). As the bacteriophage host, *E. coli*, replicates every ~30 min, the effect of the infection can be seen after 24 h from the time of incubation. This is why we chose T4 bacteriophage as a model virus. As the world is still handling the COVID-19 pandemic, and SARS-CoV-2 shows prolonged survival on surfaces, we wanted to show that our coating is effective. CCV was used as a safe surrogate. 

The antiviral activity against CCV was examined using the well-known TCID_50_/mL method. This technique is based on the cytopathic effect. If the cells appear round and detached from each other, they are infected by the virus. On the other hand, if the cells are oblong-shaped and layer the surface, there is no cytopathic effect, the viruses are inactive and have not infected the cells. 

The PC/SiO_2_-urea-Cl and PC/SiO_2_-urea-Cl-thymol coatings reduced the TCID_50_/mL of CCV below the detection level (at least ~1.5 log) compared to corona-treated PC surfaces used as a control. The TCID_50_/mL value of the bare PC surfaces was 1.06 × 10^5^, and the viruses inoculated on the coated surfaces could not be detected (Figure 11A). The PC/SiO_2_-urea-thymol titer of the inoculated viruses was below the detection limit.

CCV was detected by examining the cytopathic effect of the CRFK cells. A cytopathic effect is well recognized for CRFK CCL94 cells inoculated with CCV for three hours on bare polycarbonate surfaces (Figure 11B). On both coated surfaces, there is no cytopathic effect, indicating that the coatings effectively inactivate the virus (Figure 11C,D). 

To determine which component of the coating is antiviral, plaque assays were performed with T4 bacteriophage. The number of plaques corresponds to the number of active viruses. Three types of coatings were examined: PC/SiO_2_-urea-Cl, PC/SiO_2_-urea-thymol, and PC/SiO_2_-urea-Cl-thymol. Our results indicate that PC/SiO_2_-urea-thymol enhances viral persistence on the surface, while PC/SiO_2_-urea-Cl reduces the number of bacteriophages by about 84% (Figure 12A). Surprisingly, the combination of thymol and Cl had an improved antiviral activity, as it reduced the number of viruses by four orders of magnitude. This suggests that thymol and Cl act synergistically as antiviral agents. Figure 12B,C show agar plates with a layer of *E. coli* cells infected with bacteriophages inoculated on the bare and coated surface, respectively. Each transparent spot represents a single bacteriophage. Transparent spots could not be detected on the coated surface, indicating inactivation of the bacteriophages that were inoculated on it. 

Overall, these antiviral assays show that the coating that combines thymol and chlorine exhibits excellent activity toward two rather different viruses based on both RNA and DNA. The presence of thymol improves the activity, yet it is not crucial for the performance of this coating. However, the presence of activated chlorine is essential. If the coating does not contain Cl^+^, the number of phages increases.

## 4. Summary and Discussion 

SiO_2_-urea, SiO_2_-urea-Cl, SiO_2_-urea-thymol and SiO_2_-urea-Cl-thymol thin coatings on PC films have been successfully prepared, as evidenced by AFM, FTIR/ATR, CA and XPS. The durability of these coatings was demonstrated via an adhesive tape peeling test and their long-term stability in aqueous solution. Increasing the thickness of the SiO_2_-urea-Cl coating on the PC film led to increased activated Cl^+^ content.

The release rates of Cl^+^ as well as thymol were found to be temperature-dependent. The rate is much slower at 4 °C than at room temperature. Storing the coated PC films at −4 °C completely prevented the release of both Cl^+^ and thymol.

The antiviral activity of the coatings was tested against two different viruses, T4 bacteriophage (DNA) and canine coronavirus (RNA), safely mimicking the highly significant and widely spread SARS-CoV-2. The present study illustrates that PC/SiO_2_-urea-thymol enhances viral persistence on the surface, while PC/SiO_2_-urea-Cl reduces the number of bacteriophages by about 84%.

Surprisingly, the combination of thymol and Cl showed improved activity, as it reduced the number of viruses by four orders of magnitude. This suggests that thymol and Cl act synergistically as antiviral agents. Further work examining the effect of the above thin coatings on other types of viruses, as well as other microorganisms including bacteria and fungi, are ongoing in our laboratories.

## Figures and Tables

**Figure 1 jfb-14-00270-f001:**
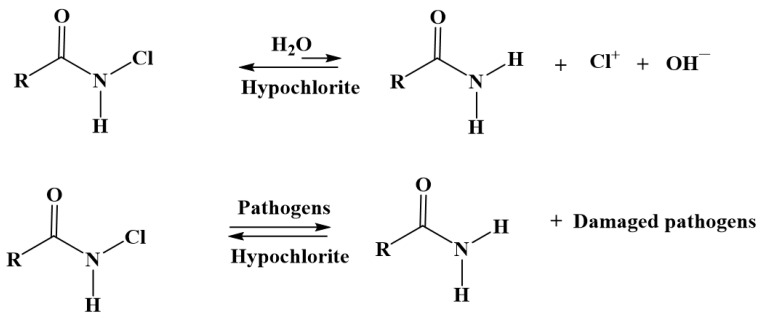
Oxidative transfer of chlorine from a chlorinated amide to (**1**) water and (**2**) pathogens.

**Figure 2 jfb-14-00270-f002:**
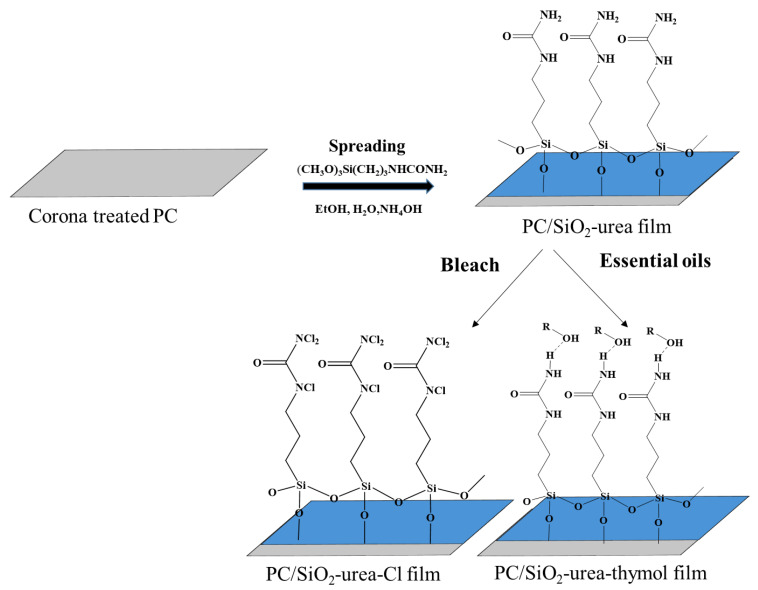
Production of SiO_2_-urea-Cl and SiO_2_-urea-thymol thin coatings on polycarbonate (PC) films.

**Figure 3 jfb-14-00270-f003:**
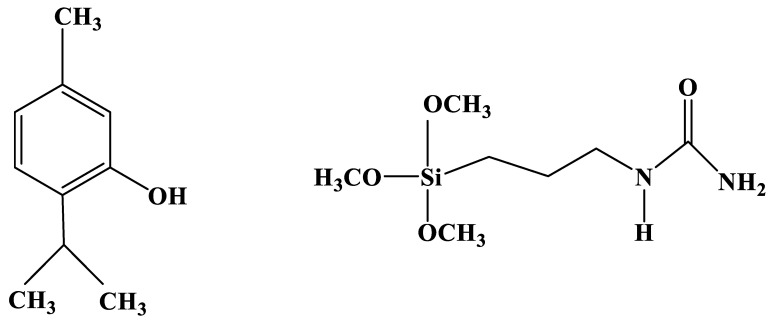
Chemical structures of thymol (**left**) and TMSPU (**right**).

**Figure 4 jfb-14-00270-f004:**
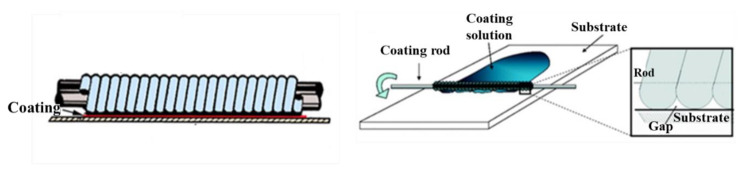
Mayer rod system.

**Figure 5 jfb-14-00270-f005:**
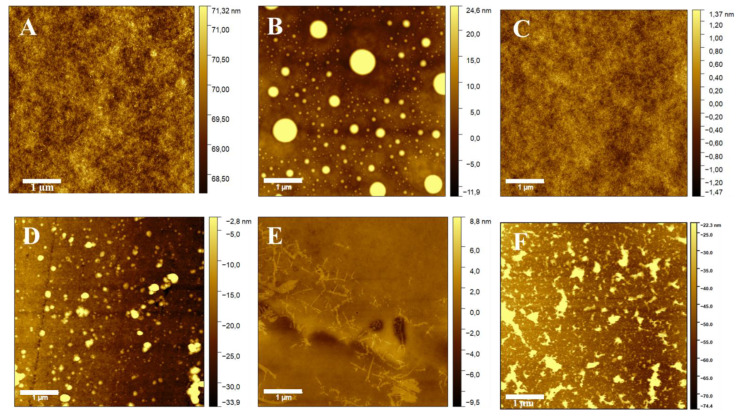
AFM topography analysis of PC (**A**), corona-treated PC (**B**), PC/SiO_2_-urea (**C**), PC/SiO_2_–urea-Cl (**D**), PC/SiO_2_–urea-thymol (**E**) and PC/SiO_2_-urea-Cl-thymol films (**F**).

**Figure 6 jfb-14-00270-f006:**
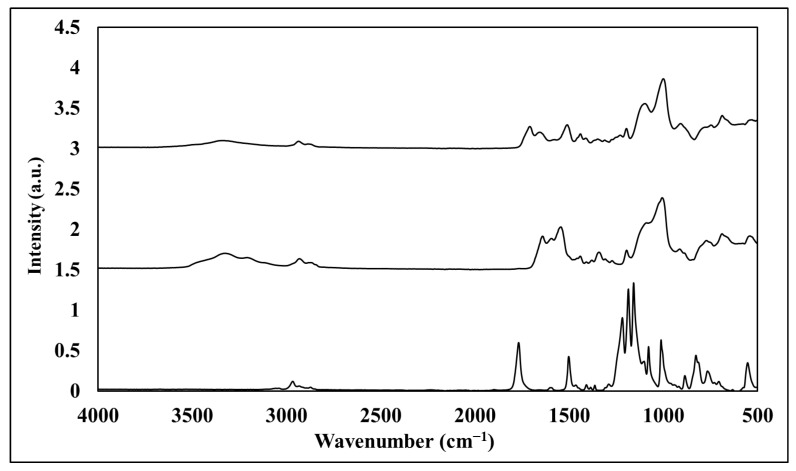
FTIR/ATR spectra of PC (bottom), PC/SiO_2_-urea (middle) and PC/SiO_2_-urea-Cl (top) films.

**Figure 7 jfb-14-00270-f007:**
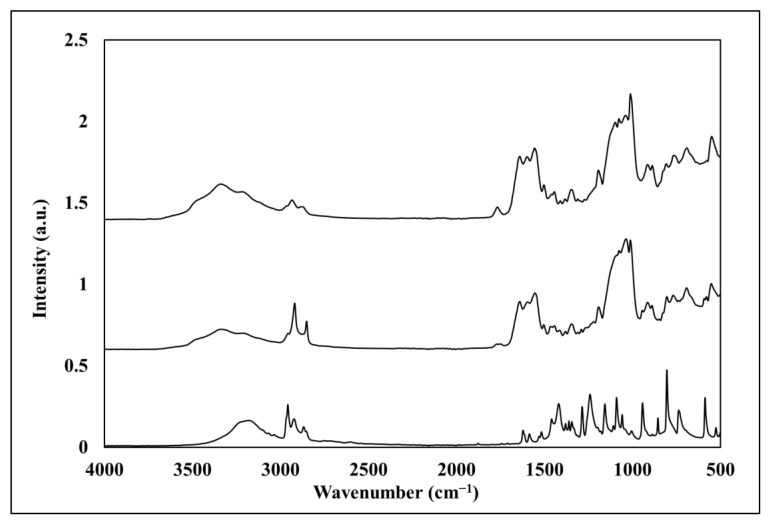
FTIR/ATR spectra of thymol (bottom), PC/SiO_2_-urea-thymol (middle) and PC/SiO_2_-urea-Cl-thymol (top) films.

**Figure 8 jfb-14-00270-f008:**
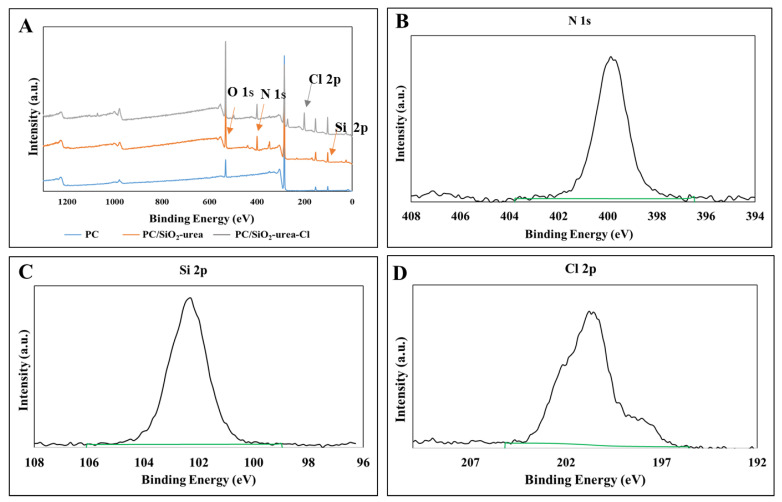
(**A**) XPS spectra of corona-treated PC (blue), PC/SiO_2_-urea (orange) and PC/SiO_2_-urea-Cl (gray) films. Magnification of the PC/SiO_2_-urea film spectra depicts the peaks corresponding to (**B**) N 1s and (**C**) Si 2p, belonging to the SiO_2_–urea coating. Magnification of the PC/SiO_2_-urea-Cl film spectrum shows the peaks corresponding to (**D**) Cl 2p, belonging to SiO_2_-urea-Cl coating.

**Figure 9 jfb-14-00270-f009:**
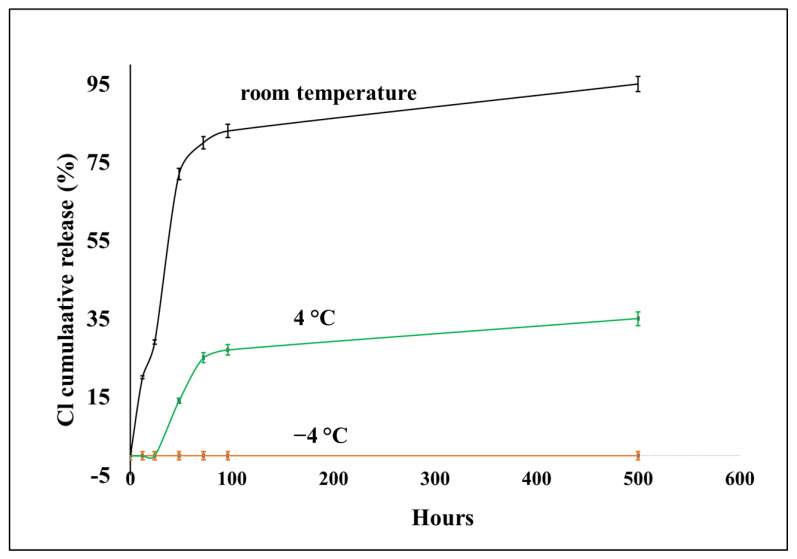
Release rates of activated chlorine from PC/SiO_2_-urea-Cl film at room temperature, 4 °C and −4 °C.

**Figure 10 jfb-14-00270-f010:**
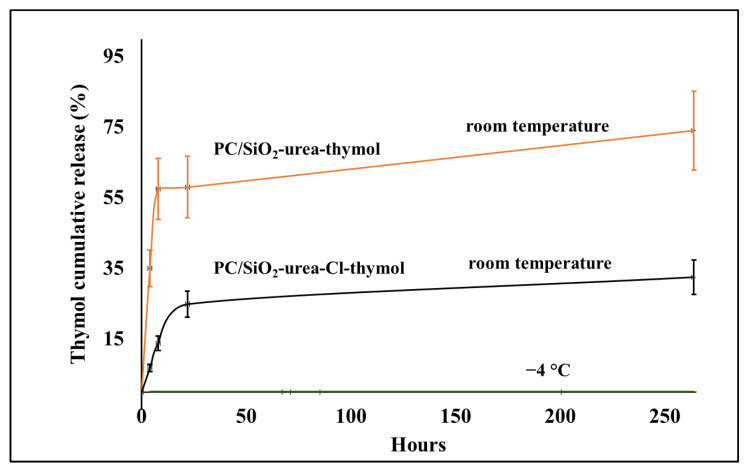
Release rates of thymol from PC/SiO_2_-urea films at room temperature and −4 °C.

**Figure 11 jfb-14-00270-f011:**
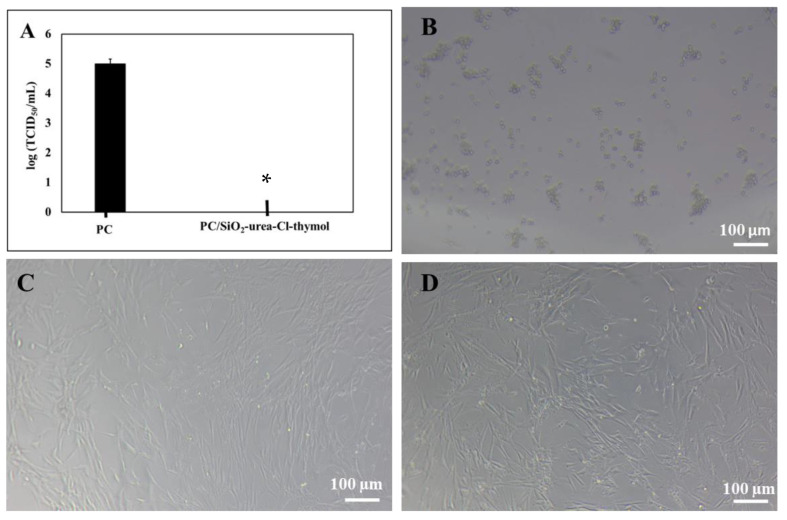
Antiviral activity against CCV (**A**) showing log(TCID_50_/mL) on the bare PC and SiO_2_-urea-Cl-thymol coated surfaces. Microscope images ×100 of CRFK cells incubated with CCV on bare PC (**B**) and on PC coated with SiO_2_-urea-Cl (**C**) or SiO_2_-urea-Cl-thymol (**D**). The SD is based on one experiment in triplicate. * represents Student’s *t*-test at a significance level of *p* < 0.05.

**Figure 12 jfb-14-00270-f012:**
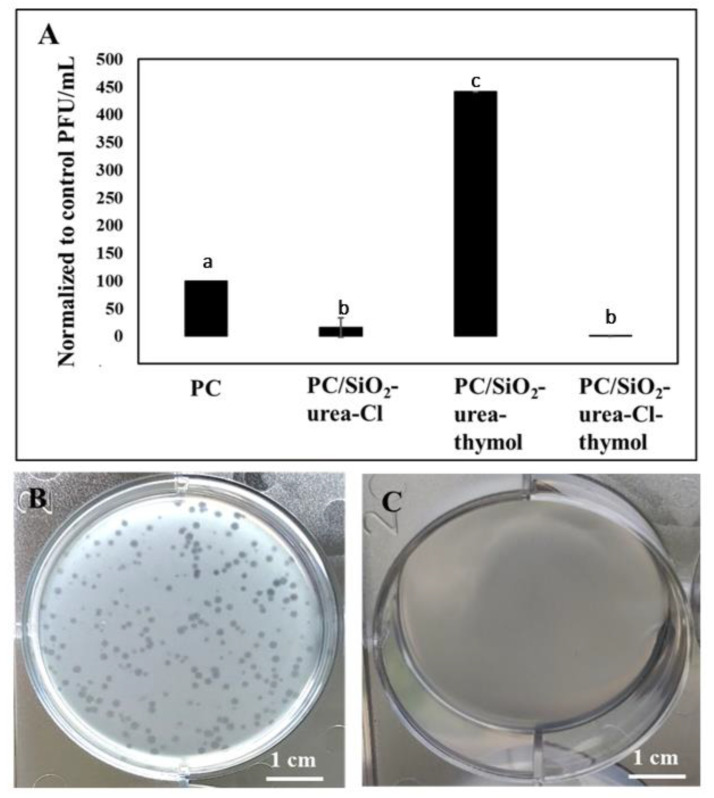
Antiviral activity against bacteriophage T4. (**A**) Normalized PFU/mL on bare PC and coated surfaces. Images show a layer of *E. coli* on wells with 1.75 cm radius after seeding bacteriophages from (**B**) bare PC and (**C**) PC/SiO_2_-urea-Cl-thymol. The SD is based on three experiments performed in triplicate. Each letter represents an insignificant group, using an ANOVA test and Fisher’s least significant difference (LSD) post hoc test, with a significance level of *p* < 0.05.

**Table 1 jfb-14-00270-t001:** Surface roughness of the various films.

Film ^a^	Roughness (nm) ^b^
PC	0.40 ± 0.02
Corona-treated PC	11.7 ± 0.5
PC/SiO_2_-urea	0.85 ± 0.01
PC/SiO_2_-urea-Cl	6.7 ± 0.3
PC/SiO_2_-urea-thymol	1.2 ± 0.1
PC/SiO_2_-urea-Cl-thymol	9.1 ± 0.7

^a^ PC films were corona-treated and then coated with TMSPU using a rod-coating technique, followed by chlorination or thymol interaction. ^b^ Average of three areas of the film (three measurements per area).

**Table 2 jfb-14-00270-t002:** Water CAs of PC and corona-treated PC and PC/SiO_2_-urea films.

Film ^a^	CA (°)
PC	74 ± 2
Corona-treated PC	40 ± 2
PC/SiO_2_-urea	69 ± 3
PC/SiO_2_-urea-Cl	93 ± 4
PC/SiO_2_-urea-thymol	110 ± 6
PC/SiO_2_-urea-Cl-thymol	102 ± 4

^a^ Corona-treated PC films were coated according to the experimental procedure.

**Table 3 jfb-14-00270-t003:** Effect of SiO_2_-urea-Cl coating thickness on the concentration of Cl bonded to the coating.

Wet Film Thickness ^a^ (µm)	[Cl] (µmol/cm^2^)
120	0.75
200	1.5
400	3.5

^a^ Corona-treated PC films were coated as described above and chlorinated with 0.5% bleach (12 h, 40 °C).

## Data Availability

Raw/processed data cannot be shared at this time due to technical and time limitations.

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
