# Peer review of "Engineered Cross-Linked Silane with Urea Polymer Thin Durable Coatings onto Polymeric Films for Controlled Antiviral Release of Activated Chlorine and Essential Oils"

_jfb, 2023, doi:10.3390/jfb14050270_

Round 1
Reviewer 1 Report
This manuscript described the synthesis and characterization of urea polymer thin durable coatings onto polymeric films for controlled antiviral release of activated chlorine and essential oils. The manuscript was well described and organized for the synthesis of PC/SiO2-urea-thymol and PC/SiO2-urea-Cl-thymol and for the release rate of thymol and CCV or Bacteriophage assay. I hope the authors can review the manuscript according to the following questions and comments.
1. The authors explained " the combination of thymol and Cl had an improved antiviral activity as it reduced the amount of viruses by four orders of magnitude" in Fig.12. Please explain a little bit more details for that reason.
2. The authors described the release from PC/SiO2-urea-Cl-thymol is considerably slower 317 (~25%). This can be due to the hydrophobic physical interaction between the Cl of the N halamine and thymol. I am just wondering how the authors verified the hydrophobic interaction.
3. I am just wondering that the authors checked the release rate for the different Cl content samples. Please explain the correlation for that.
Reviewer 2 Report
In this work, authors have prepared a series of cross-linked coatings of SiO2 with urea, urea-Cl, urea-thymol and C-thymol and have dully characterized them using various techniques. The coatings were tested against two different viruses and their antiviral properties were thoroughly evaluated. Authors have neatly planned and executed the research project. Article is presented nicely, narration is smooth and English used is simple and understandable. All the claims of the authors are well supported by the experimental data and references.
Reviewer 3 Report
Sasson et al., prepared polymeric films with antiviral activities based on chlorine and thymol, where the crosslinking took place using SiO2 and urea. They aimed to control the release of antiviral agents, where they found that the release persists even after 22 h (60% release). The antiviral activities have been carried out against CCV, where the Cl-thymol contributes to the antiviral activities. However, there are some concerns which authors should be addressed before the manuscript could be considered further.
Please see my comments below:
ABSTRACT
1. Line 16 “..and together.” Consider and ‘its combination’
2. Optimized preparation parameters should be stated in the abstract.
3. Please state how exactly the release was controlled.
INTRODUCTION
4. Line 32. “…4.3 million confirmed deaths” as of?
5. In the first paragraph, please incorporate this research on how the spotlight is given to materials with anti-SARS-CoV-2 properties (Chiari et al., Polymers 2022, 14(16), 3297; doi: 10.3390/polym14163297).
6. Moreover, to increase the research urgency please include how COVID-19 pandemic is still relevant due to limited FDA-approved anti-SARS-CoV-2 drugs (Sharun et al., Narra J 2022, 2(3), e92)
7. Line 61—65 it is of importance to highlight the state of the art of thymol as anti-SARS-CoV-2. I suggest author to read and incorporate this study Iqhrammullah et al., Scientia Pharmaceutica 2023, 91(1), 15; doi: 10.3390/scipharm91010015. Please focus on this sub-heading “5.3. Other In Vitro Studies on Anti-SARS-CoV-2 Activity of Essential Oils”
8. There is a little evidence for thymol as anti-SARS-CoV-2, authors may argue its availability as the advantage of using this oil.
9. In the last paragraph of introduction, authors did a well elaboration on the study rationales. However, it is better to improve the novelty statement in the introduction. Please compare to published studies reporting materials with antiviral activities.
11. In Figure 2. What is R in the PC/SiO2-urea-thymol film?
METHODS
12. Line 95. “Corona-treated PC films” what do you mean by Corona treated PC film?
13. Line 97. “Escherichia coli” should be written as Italic. Please avoid this mistake in the other part of the manuscript.
14. In characterization, please disclose if the samples were pre-treated or not.
15. Wondering why authors did not measure the effect of time to the antiviral activities, as the main concern of the study is about the release control.
RESULTS AND DISCUSSION
16. “This spectrum shows that thymol is present….” Not only present, authors should be able to interpret the possible interactions occurring between the active ingredients and the material.
17. Is it possible to determine the release kinetics?
18. Please perform statistical analysis for the antiviral activities.
CONCLUSION
19. I suggest author provide “Conclusion” instead of “Summary”. Consequently, authors should revise their writing by focusing more on the results and their implications.
20. “Thin cross-linked coatings of SiO2-urea, SiO2-urea-Cl, SiO2-urea-thymol and SiO2-urea-Cl-thymol were prepared onto PC films” I don’t think the statement is suitable as concluding remarks. Author may say “…the SiO2-urea-Cl-thymol has been successfully prepared, evidenced by the characterizations..” Please adjust.
21. “The chlorine and thymol bound to the coatings were quantified, and the release rates were studied at different temperatures.” Experimental statement like this should be avoided in the conclusion.
22. Authors should highlight how the release of the antiviral agents could be controlled as this is what claimed by the authors in the title. Also please do tell, how the control could implicate the efficacy of the antiviral material.
Reviewer 4 Report
The manuscript discusses the interesting topic and research findings are good to be considered for practical applications. I would like to suggest the following modifications:
1. Abstract needs to write in more descent way to attract the readers and interesting. For example, the sentence, "Here, the preparation and characterization of new antiviral coatings onto polycarbonate (PC) for controlled release of activated chlorine (Cl+ ) and thymol separately and together"
Please improve the writing throughout the manuscript.
2. Introduction: The last paragraph is not complete and feels like there is not motivation behind the work proposed. It should be clearly stated.
3. Section 2.2.4: all the subsection must be numbered because it is not very clear about the techniques.
4. Section 3.3.1: how the error in roughness is calculated ?
5. Figure 11 should be explained very well. which micrograph it is and how it is done ?
6.
Round 2
Reviewer 3 Report
Authors have made necessary revisions and responded to may concerns. The submitted manuscript is now can be accepted.
Reviewer 4 Report
Authors have responded well and further modification is not required.
'